# PathUp: Patch-wise Timestep Tracking for Multi-class Large Pathology Image Synthesising Diffusion Model

## ABSTRACT

In digital pathology, cancer lesions are identified by analyzing the spatial context within pathology images. Synthesizing such complex spatial context is challenging as pathology whole slide images typically exhibit high resolution, low inter-class variety, and are sparsely labeled. To address these challenges, we propose PathUp, a novel diffusion model tailored for the synthesis of multi-class high-resolution pathology images. Our approach includes a latent space patch-wise timestep tracking, which helps to generate high-quality images without tiling artifacts. Expert pathology knowledge is integrated into the model through our patho-align mechanism. To ensure robust generation of lesion subtypes and scale information, we introduce a feature entropy loss function. We substantiate the effectiveness of our method through both qualitative and quantitative evaluations, supplemented by assessments from human experts, demonstrating the authenticity of the synthetic data produced. Furthermore, we highlight the potential utility of our generated images as an augmentation method, thereby enhancing the performance of downstream tasks such as cancer subtype classification.

## CCS CONCEPTS

• **Computing methodologies → Reconstruction**; *Image representations*; *Information extraction*; **Visual content-based indexing and retrieval**.

## KEYWORDS

Image Synthesis, Diffusion Model, Cross-Modality Knowledge Alignment, Digital Pathology

## 1 INTRODUCTION

Histopathology involves diagnosing and studying diseases by examining histology images collected under a microscope [10, 39, 40]. Histology images of tissue contains both complex and ambiguous information, challenging pathologists to perform a robust, reproducible and efficient analysis. Thanks to the advances in Deep Learning (DL), impressive performance have been witnessed in various digital pathology tasks, including cancer classification and grading [49, 54], cell detection and segmentation [36, 46], interpretation of multiplex immunohistochemistry [19, 45], etc.

The superiority of DL-based digital pathology analysis comes at a cost of acquiring large, high-quality annotated training datasets.

*ACM MM, 2024, Melbourne, Australia*
© 2024 Copyright held by the owner/author(s). Publication rights licensed to ACM.
ACM ISBN 978-x-xxxx-xxxx-x/YY/MM
https://doi.org/10.1145/nnnnnnn.nnnnnnn

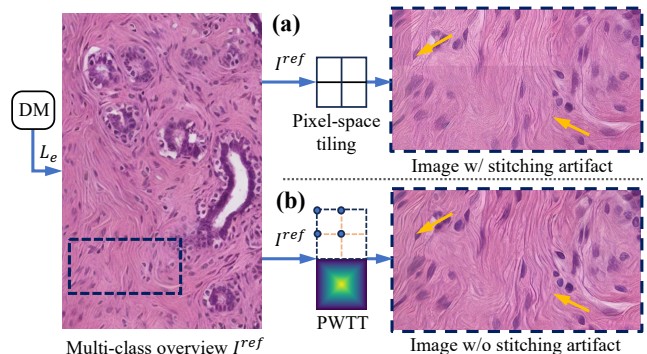

**Figure 1: Our proposed PathUp trains a diffusion model guided by our feature entropy loss $L_e$ to do both low resolution overview $I^{ref}$ generation and high resolution pathology image synthesis. Comparing with pixel space tiling (a) which has sharp tiling edges, our proposed latent space patch-wise timestep tracking method (b) generates high-resolution image with smooth transition.**

However, the available annotated images are still scarce when it comes to various lesion subtypes driven by different microenvironment and multiple biological factors, or scale-variable regions with discriminative morphological patterns. The limitation of training data drawbacks the prediction performance of learning algorithms. To this end, one solution is to train a generative model that can produce realistic pathology images that augments existing data. Generative models have been proposed to help learning methods in various tasks such as nuclei segmentation [31, 38], survival prediction [6, 17] and cancer grade estimation [14, 52].

The synthesis of high-resolution pathology images typically contains two principal stages: (1) creating of class-specific layout images, and (2), incorporating high-resolution features under the guidance of the layout image, with an effort to remove tiling artifacts. However, existing methodologies often struggle to achieve both of these aforementioned stages. Certain approaches focus on stage (2), producing detailed representations in small patches through the utilization of either randomized or predetermined layouts [1, 4], or alternatively, they focus on the generation of giga-pixel Whole Slide Images (WSIs) devoid of class-specific conditions [3, 20]. The challenge of tiling artifacts has been addressed through the introduction of consistent loss functions for the generated images [30], or by employing pixel-space shifting windows [20]. However, these methodologies miss the opportunity to learn the abundant spatial context inherent in heterogeneous lesions ranging over varying resolutions, consequently losing diagnostically crucial information relevant to cancer biology. Furthermore, the approaches to tiling artifact removal predominantly focus on imposing constraints or

tiling images in pixel space, which may be challenging for images featuring multi-class subtypes of lesions.

Spatial context in pathology images includes how different types of tissue distributed around each other, as well as how they form architectural patterns that supports lesion classification and diagnosis (e.g. normal tissue, pathological benign, invasive carcinoma, etc.). Plenty of evidence have demonstrate the importance of spatial context in cancer diagnosis and prognosis [10, 39]. For example, Invasive Papillary Carcinoma (IPC) (i.e. cancer cells moving into nearby tissue), is a biomarker associated with an increased risk of lymph node metastasis in breast carcinoma, usually diagnosed by finding predominantly papillary architecture [40].

Given the biological significance of architectural spatial context within pathology, we hypothesize that generating high-resolution pathology images with meaningful architectural lesion patterns holds significant potential to enhance various downstream tasks. The most challenging task we resolve is modeling complex spatial contexts utilizing limited information while seamlessly eradicating tiling artifacts through the employment of a latent space timestep tracking strategy. To capture the spatial contexts, we advocate the adoption of diffusion model as a robust solution for synthesizing high-resolution pathology images devoid of tiling artifacts. Formally, we introduce the patho-align module, which integrates multi-resolution pathological knowledge into a novel latent diffusion model [41]. This model facilitates the generation of multi-class spatial lesion contexts across various resolution levels. To ensure robust generation, we introduce an feature entropy loss function for patho-align, aiming at minimizing inter-prompt distances while simultaneously maximizing intra-prompt distances.We then bridge resolution disparities through a timestep tracking strategy operating within the latent space, achieving the generation of high-resolution images by aggregating low-resolution latent patches. Leveraging a latent weight map, we effectively mitigate tiling artifacts without additional postprocessing methods. With the help of a latent weight map, we remove the tiling artifacts without adding any other postprocessing methods.

Fig.1 illustrates the image generation procedure highlights the efficacy of our method in eliminating tiling artifacts. Notably, the synthetic image not only replicates the layout observed in the low-resolution reference image but also exhibits seamless transitions along the edges of each patch. In the experiment section, extensive analyses are presented to substantiate the advantages afforded by our approach. Furthermore, we showcase the utility of augmented images generated by our model in training downstream tasks, such as lesion subtype classification.

To summarize, our contributions are as follows:

- We propose the first generative model to learn the generation of multi-resolution lesion subtypes from pathology images.
- We introduce patho-align, which incorporates expert pathology knowledge with multi-class images. A feature entropy loss function is proposed to increase the inter-class variety for synthetic images.
- We present a patch-wise timestep tracking strategy that within the latent diffusion model framework. This strategy enhances the model's capacity to generate high-resolution

images, and concurrently utilizes the latent weights to address tiling artifacts.
- We show that our method is capable to generate realistic pathology image in different resolution. The synthetic pathology images can be used as a data augmentation method, and we demonstrate the efficacy of the augmentation data in downstream tasks such as lesion subtype classification.

We stress that the benefit of modeling multi-resolution spatial context is beyond data augmentation. This topic improves the understanding and quantifying of the architectural patterns of tumor microenvironment, and provides a foundation for correlating spatial context with genomics and clinical outcomes. Such direction is where we step towards.

## 2 RELATED WORK

### 2.1 Generative Modeling for Pathology Images

Pathology image generation has been the subject of extensive investigation. Some of the explored methodologies rely on texture-based image synthesis techniques [15, 21]. However, such methods often encounter challenges related to their limited generalizability. In contrast, DL-based approaches for image generation leverage the capacity to acquire complex patterns from large-scale training datasets, thereby enabling the generation of diverse and realistic images. This capability has been underscored by several studies [11, 32, 53] utilizing Generative Adversarial Networks (GANs) [18]. Notably, however, these methods focused on generating low-resolution patches rather than high-resolution images, and suffered from instability and mode collapse issues [33, 35].

Recently, diffusion models have gained popularity in medical image synthesis, demonstrated superior performance over GANs [20]. However, generating high-resolution images using diffusion models poses a significant computational challenge due to the escalating computational costs associated with increasing image resolution. One strategy to tackle this challenge involves leveraging latent diffusion models (LDMs) [41]. Despite yielding impressive results, the achievable resolutions demonstrated in LDMs [5, 41] remain limited. Alternative approaches [24, 42] achieve high-resolution image generation by cascading a series of upscaling diffusion models. Our model involves only a single upscaling phase comparing with these alternative approaches. Furthermore, in contrast to our work, existing methods utilizing diffusion models are confined to generating random tissue images, thereby limiting their applicability in downstream tasks.

### 2.2 Latent Similarity Estimation

In image analysis, similarity metrics are crucial in resemblance quantification. Similarity metrics aim to measure how "close" two images are. Traditional point-wise difference metrics, such as Euclidean $l2$ and Manhattan $l1$, are limited in their ability to capture joint statistical characteristics. Consequently, methodologies mimicking human visual system, such as SSIM [50], MS-SSIM [51], and FSIM [56], have been developed. While effective in scenarios where structural ambiguity is minimal, these methods may fall in tasks where synthesizing complex structures is crucial. such as in text-to-image generation tasks.

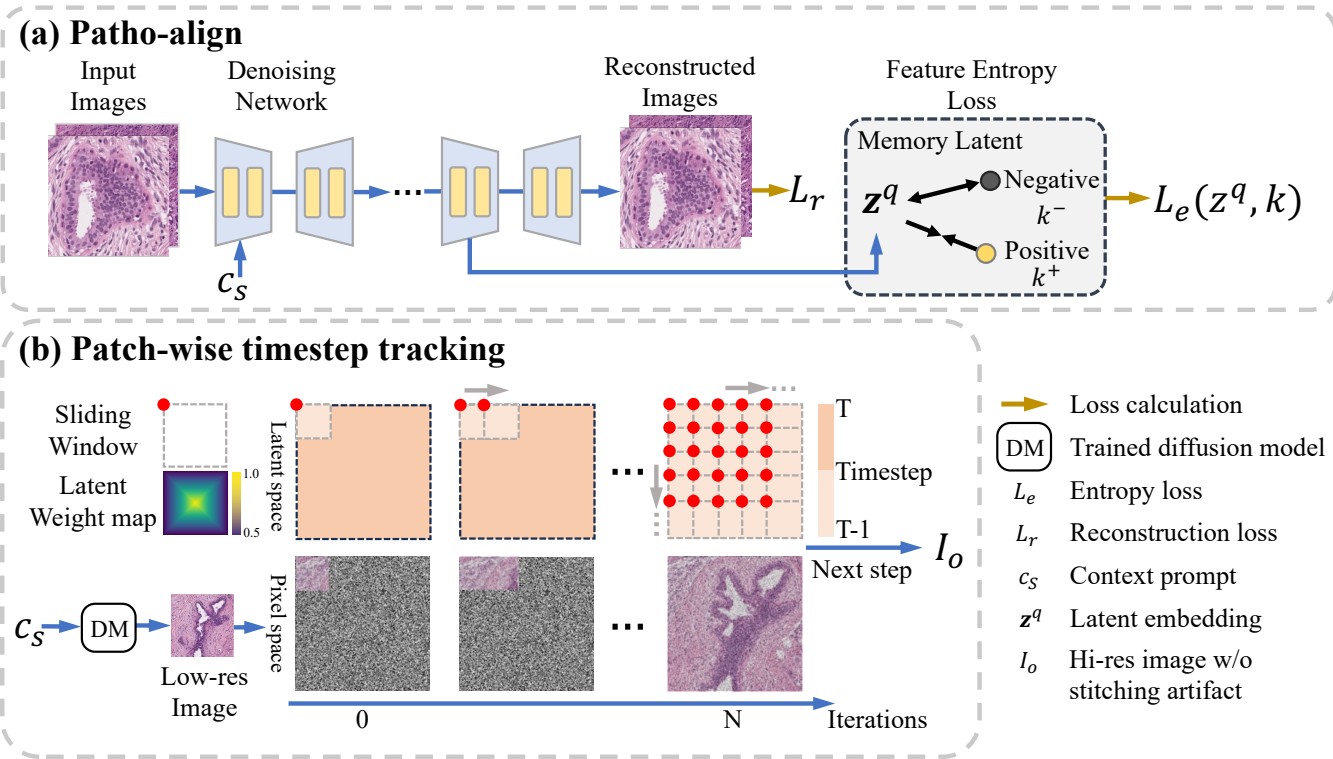

**Figure 2: Our proposed PathUp framework has two key components: (a) The Patho-align module, which integrates multi-class pathology images at various spatial levels along with textural descriptions $c_s$ into the latent diffusion model. Training is guided by a feature entropy loss, which leverages a memory latent to ensure that latents $z^q$ from the same class exhibit closer distances. (b) To facilitate the generation of high-resolution images from low-resolution references without requiring additional training, we propose a patch-wise timestep tracking module. This module operates by individually denoising split latent patches and simultaneously removing tiling artifacts through the utilization of a latent weight map.**

Recent advancements in computer vision have delved into methodologies for assessing similarity within the latent space of deep neural networks, commonly denoted as 'perceptual loss' or 'feature matching loss' [13, 26, 47], which have exhibited notable improvements, particularly in image synthesis contexts [2]. However, it is notable that these techniques often rely on pretrained backbone networks trained on datasets dissimilar to pathology images. Consequently, we aim to explore the potential of leveraging latent codes acquired through Latent Diffusion Models (LDMs) for the synthesis of medical images.

Estimating similarity between latent representations holds importance in contrastive learning approaches [9, 23, 27, 48]. Typically, these approaches optimize a loss function tailored to minimize the feature distance between positive target instances while concurrently maximizing it against a set of negative targets. Drawing inspiration from these endeavors, our objective is to harness feature distances for medical image synthesis employing diffusion models. To achieve this, our approach combines texture features and visual features into a unified query, while maintaining a memory latent as a comparison target. This strategy enables us to effectively leverage both texture and visual information, thereby enhancing the synthesis process by ensuring closer distance between latent

representations corresponding to similar instances and maximizing their distance from dissimilar instances.

## 3 METHOD

Considering an authentic pathology image, the spatial arrangement formed by the collection of tissues and cells serves as a biomarker for tumor classification. Motivated by this, we propose our pathology image synthesis pipeline. Our approach involves patho-align mechanism, which integrates multi-class pathology images at diverse resolutions alongside textural prompts into the diffusion model. To ensure robust generation of texturally relevant features, we introduce a novel feature entropy loss. For the synthesis of high-resolution images, we adopt a strategy that splits the latent code into overlapping tiles and deploy patch-wise timestep tracking. As a result, our methodology alleviates tiling artifacts and effectively bridges the gap between multi-resolution images.

### 3.1 Patho-align

Crafted to synthesize spatial layouts across various scales and classes, our patho-align module is tailored to leverage pathological knowledge from multiple scale pathology images, thereby generating class-correlated spatial contexts across multiple scales. We

---

**Algorithm 1** Whole PathUp Inference Logic

---

1: **Input**: Low resolution synthetic reference overview $I^{ref}$, textural guidance $c_s$
2: **Parameter**: Latent patch size $p$, overlap pixels $o$, patch latent weight $w$
3: **Output**: High-resolution image $I^h$
4: $X_0 \leftarrow \mathcal{E}(I^{ref})$
5: $X_t \leftarrow \sqrt{\alpha_t}X_0 + (1 - \alpha_t)\,w$
6: Split $X_t$ into $N$ patches according to $p$, $o$
7: **for** Timestep $t$ in $[T, T-1, ..., 0]$ **do**
8:    **for** Latent patch $x_t^n$ in $[1, 2, ..., N]$ **do**
9:       $x_{t-1}^n \leftarrow \hat{d}\left(x_t^n, c_s\right)$
10:    **end for**
11:    Combine $x^n$ according to $w$ for $X_{t-1}$
12: **end for**
13: **return** High resolution synthetic image $I^h \leftarrow \mathcal{D}(X_0)$

---

achieve this objective by introducing a patho-align strategy and a feature entropy loss for training a latent diffusion model using sparsely labeled pathology images.

In the context of a latent diffusion model, input image $I$ is fed into a predefined encoder $\mathcal{E}$ to create a embedding $x_0 = \mathcal{E}(I)$, upon which the diffusion process is applied. Subsequently, a decoder $\mathcal{D}$ reversely projects the latent back to the pixel space, ensuring fidelity with the original image $I$. The noise is gradually injected into the latent variable $x$ occurs over $t = 1 \ldots T$ using a steps via a Markovian forward process, expressed as:

$$x_t = \sqrt{\alpha_t}x_0 + (1 - \alpha_t)\,w \qquad (1)$$

here, $x_t$ represents the latent variable at step $t$, $w \sim \mathcal{N}(\mathbf{0}, \mathbf{I})$ denotes a noise term, and $\alpha t$ controls the noise schedule. Treating the diffusion model $\hat{x}$ as an optimization problem, its loss can be defined as:

$$L_r := \mathbb{E}_{x, c_s, t}\left[\sigma_t\,\|\hat{x}\,(x_t, c_s) - x_0\|_2^2\right] \qquad (2)$$

where $\sigma_t$ is a noise schedule term, $\hat{x}(\cdot, \cdot)$ denotes the image generation process of a text-guided diffusion model, $c_s$ serves as a conditioning vector providing textual guidance.

To facilitate training of the multi-resolution pathology image generator, uniformed image $x$ paired with its corresponding text $c_s$ is required. However, due to the substantial expertise and associated costs of pathologists, furnishing detailed image descriptions for every individual image is impractical. Consequently, for pathology images characterized by non-uniform resolutions and limited image descriptions, we propose a data preparation protocol. This protocol involves utilizing class information from each image to generate a prompt string $c_s$, with spatial levels such as 'overall' and 'patch' added individually. While this protocol ensures a prompt for each image, the scarcity of $c_s$ may pose a potential risk in generating images with low inter-class variety.

## 3.2 Feature Entropy Loss

In addressing the challenge of inter-class variety while maintaining intra-class generation performance, we propose a Feature Entropy Loss (FEL) as a robust mechanism to learn from sparsely labeled

pathology images. Inspired by contemporary findings, we characterize the distribution of training images as evidence of sampled inter-class variation. The objective of our loss function is to ensure that images sharing the same prompt $c_s$ exhibit high representation similarity compared to those with different prompts.

To achieve this, we maintain a memory latent for the combined texture-vision features to reduce the distance between images with the same prompt while increasing the disparity between images with different prompts. We employ a modified cross-entropy formulation to accomplish this objective, which mathematically takes the form:

$$L_e = \mathbb{E}_{z, k}\left[-\log \frac{\exp\left(z \cdot k^+/\tau\right)}{\sum_{i=0}^{K}\exp\left(z \cdot k_i^{K-1}/\tau\right)}\right] \qquad (3)$$

where $z = e(x_t, c_s)$ represents the middle block latent generated by the encoder $e$ of the denoising U-net, $k$ denotes a $K$-dim memory latent serving as a comparison target for each possible prompt corresponding to pathology images, and $\tau$ signifies a temperature constant. When the FEL is established, the positive representation $k^+$ corresponds to the vector in $k$ that shares the same prompt as $z$, while the negative representations $k^-$ represent other vectors with different prompts. After each training step, $k$ is updated individually using $k_{n+1} = \alpha k_{n-1} + (1-\alpha)k_n$, which implements a moving average of image embeddings to introduce variance to the comparison target and prevent overfitting.

Both $L_r$ and $L_e$ are utilized to train our generator, yielding the overall loss formulation:

$$L_{LDM} = L_r + \beta L_e \qquad (4)$$

where $L_{LDM}$ denotes the comprehensive loss function employed for optimizing the LDM. By implementing this training scheme alongside the FEL, we train a generator equipped with multi-scale pathology knowledge, thereby enhancing the generation of high-resolution images. Refer to Fig. 2(a) for an illustration.

## 3.3 Patch-wise Timestep Tracking

In addressing the challenge of generating high-resolution pathology images while leveraging multi-resolution expert knowledge within the LDM, we encounter the demand for substantial computational resources. To mitigate this, we propose a patch-wise timestep tracking method aimed at reducing the computational cost while keeping the quality of generation. The detailed framework of our method is depicted in Fig. 2 (b). Importantly, our approach operates solely during the inference period with no additional training.

During inference, we partition the latent code of a synthetic reference overview image into latent patches $x^l$, each assigned an independent scheduler. These latent patches are then processed by the denoising procedure based on the timestep $t$. The denoising step for each latent patch is represented as:

$$x_{t-1}^l = \hat{d}\left(x_t^l, c_s\right) \qquad (5)$$

Here, $\hat{d}$ denotes our denoising process of LDM trained by our patho-align framework, while $c_s$ represents the textural guidance. As the denoising process operates on one latent patch at a time, the timestep across the entire image may become uneven. To address

this, for timestep $T$, we sequentially denoise all the latent patches and update timestep $T \leftarrow T - 1$. This method is illustrated in Fig. 2 (b). The latent patches are tiled to create the high-resolution latent for the subsequent timestep.

To ensure a smooth transition between overlapping tiles and mitigate tiling artifacts, we weight the latent vectors in the tiles based on their distance from the center of the tile. The weight assigned to a latent vector is computed using the following formula:

$$w = \frac{min(|p - p'|, |q - q'|)}{L_p} \quad (6)$$

where $p', q'$ denote the center of latent patch in each direction, and $L_p$ represents the width of a single latent patch, thereby normalizing the weight tile within the range $[0.5, 1]$. The resulting weight map for each tile is visualized in Figure 2 (b). Subsequently, to prevent tiling artifacts from affecting the generation, the final value of a latent vector in a target coordinate is calculated by summing all inference values of the latent vector and dividing by the sum of weights. The efficacy of these tiling strategies can be observed in Figure 4.

The inference logic of PathUp is demonstrated in Alg. 1. Initially, a synthetic pathology overview image $I^{ref}$ is the spatial context reference input for our diffusion model trained using pathology knowledge. Subsequently, a certain amount of noise, $\delta_t$, is injected into the reference image to create a noised reference image latent $X$. Following the resizing and partitioning of $X$ into $N$ latent patches, The latent tiles are then combined using weigt $w$ to generate a tiling artifact free synthetic high-resolution pathology tissue $I^h$ with cancer-related spatial context.

In summary, our proposed pipeline enables the generation of multi-resolution pathology images. In the subsequent section, we conduct extensive experiments to evaluate the effectiveness of our generated images.

## 4 EXPERIMENTS

### 4.1 Dataset

We assess the performance of our method using the publicly available BRACS dataset [8], comprising pathology images related to breast cancer extracted from 547 Whole Slide Images (WSIs). The dataset contains 4539 Regions of Interest (RoIs), each annotated with one of seven cancer subtypes: Normal (N), Pathological Benign (PB), Usual Ductal Hyperplasia (UDH), Flat Epithelial Atypia (FEA), Atypical Ductal Hyperplasia (ADH), Ductal Carcinoma in Situ (DCIS), and Invasive Carcinoma (IC). For training our generator, we divide the RoIs into $512 \times 512$ patches with a 64-pixel overlap. To address potential issues related to unbalanced data distribution, we limit the patch-level training data to 6000 patches per class. Additionally, we extract overview-level data by segmenting RoIs into large $2048 \times 2048$ patches with a 256-pixel overlap, which are subsequently resized to $512 \times 512$ dimensions.

### 4.2 Implementation Details

All experiments are conducted utilizing a Nvidia A100 GPU. During the training of our patho-align module, we employ a learning rate of $5e^{-6}$ in conjunction with the AdamW optimizer [34], spanning 50,000 iterations with a batch size of 4. We utilize the DDIM [44] noise scheduler for this process. For the feature entropy loss, $k$ is randomly initialized by 14 anchors, derived from the product of the number of cancer subtypes and the number of spatial levels. We set $\beta = 0.1$ for $L_{LDM}$. During inference, we adopt patch-wise timestep tracking, dividing the latent space into $64 \times 64$ patches with 32 overlapping to generate high-resolution images of $2048 \times 2048$ pixels.

### 4.3 Metrics

We employ a range of data assessment methods to evaluate the fidelity of synthetic pathology images. Adopting metrics from the natural image community, we incorporate qualitative and quantitative assessments tailored to the medical context. To evaluate the fidelity of synthetic images at a resolution of $512 \times 512$, we compute Improved Precision (IP) and Improved Recall (IR) metrics between real and synthetic images [29]. IP assesses synthetic data quality, while IR measures data coverage. Additionally, we conduct similarity evaluations between synthetic and real images using Fréchet Inception Distance (FID) [22] and Kernel Inception Distance (KID) [7], as suggested in prior studies [37, 43]. We evaluate the effectiveness of the proposed modules individually and conduct 5 individual generations for each experiment, calculating the standard deviation to ensure the robustness of our methods. For qualitative analysis, we engage a team of pathologists to evaluate the plausibility of the synthetic images.

### 4.4 Inter-model Evaluation of Multi-resolution Pathology Image Synthesis

To demonstrate the superiority of our method in producing synthetic pathology images, we conduct inter-model comparisons using StyleGAN2 [28], VQ-GAN [16], and LDM [41]. Our evaluation of the quality of synthetic images with PathUp includes two levels: overview and patch. At the overview level, synthetic $512 \times 512$ lesion images are compared with high-resolution patches resized to the same size. At the patch level, synthetic tiles are compared with real $512 \times 512$ pathology image patches. These models are trained using identical data and are specifically designed to generate images at each level. For all models in the comparison, we generate 10,000 samples and employ metrics to compare synthetic images with the test set of the BRACS dataset.

Table 1 showcases our method's outstanding performance compared to others at both levels. When compared to real images, our method outperforms all others in terms of IP, FID, and KID, achieving values of 0.964, 45.359, and 8.139 at the overview level, and 0.955, 66.729, and 11.742 at the patch level, respectively. Notably, our method exhibits significantly higher IP compared to StyleGAN2, with improvements of 0.308 and 0.550 at the overview and patch levels, respectively. Analysis reveals that diffusion-based methods perform better across these metrics, underscoring the efficacy of the diffusion model in pathology image generation. However, compared to LDM, our method excels in FID and KID, attributable to the patho-align module, which enables our model to generate images that closely match the dispersion patterns of real pathology images. Furthermore, the high performance of IR demonstrates our method's ability to generate data covering the full dispersion of real data. This improvement may be attributed to our feature entropy

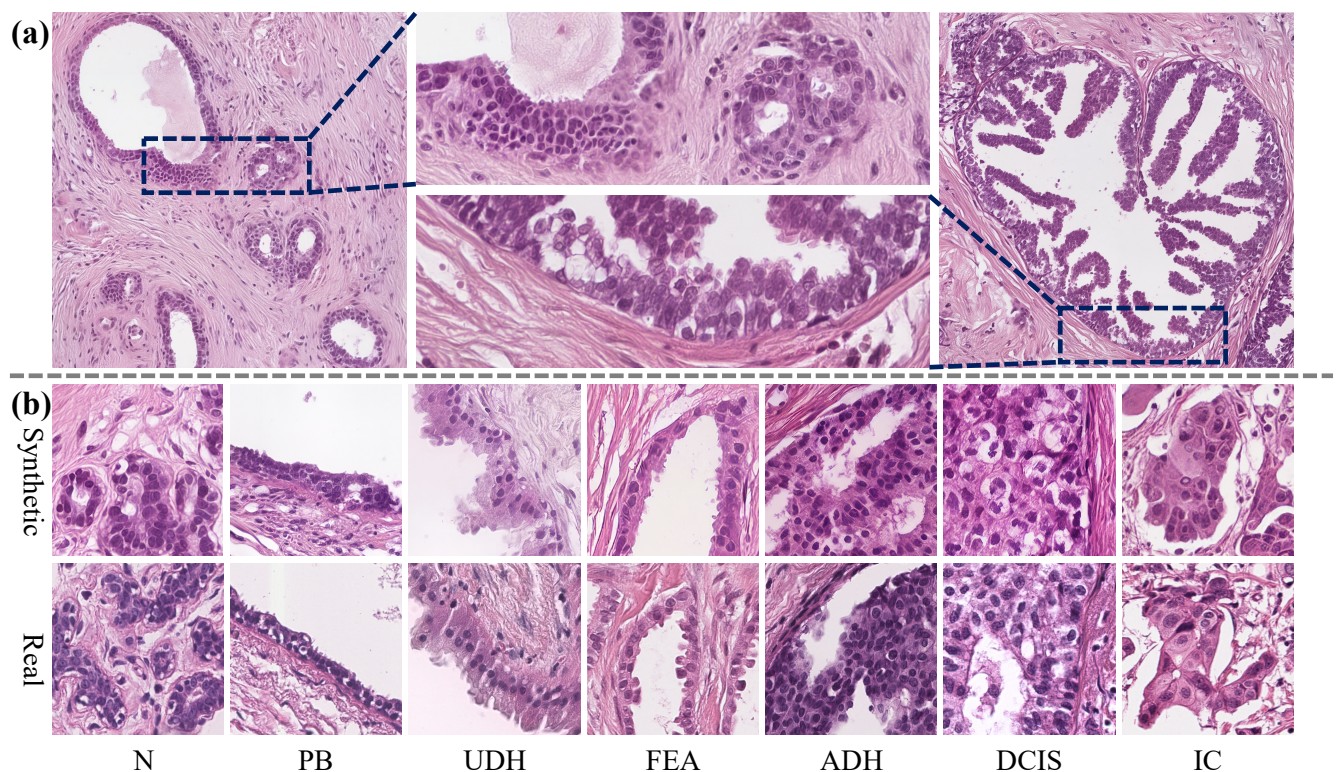

**Figure 3: Visualization of the outcomes generated by our approach. (a) Presents two synthetic high-resolution image patches: the left patch corresponds to a generation for Pathological Benign (PB), while the right patch represents a generation for Ductal Carcinoma in Situ (DCIS). (b) Demonstrates a comparison at the patch-level between our synthetic results and real patches specific to each class. Both visualizations highlight the capability of our model to generate realistic tissue images.**

**Table 1: Performance comparison between methods for pathology image synthesis. Evaluations are performed on both overview and patch level.**

|  | Models | IP↑ | IR↑ | FID↓ | KID*↓ |
|---|---|---|---|---|---|
| Overview | StyleGAN2[28] | 0.656±0.087 | 0.417±0.044 | 69.375±3.942 | 24.455±2.513 |
| | VQ-GAN [16] | 0.710±0.092 | 0.402±0.041 | 78.617±3.732 | 25.307±1.212 |
| | LDM [41] | 0.891±0.037 | 0.343±0.036 | 98.056±4.191 | 20.751±2.794 |
| | Ours | **0.964±0.012** | **0.592±0.026** | **45.359±3.732** | **8.139±0.413** |
| Patch | StyleGAN2[28] | 0.405±0.105 | 0.337±0.089 | 125.493±5.051 | 43.709±3.988 |
| | VQ-GAN [16] | 0.828±0.078 | 0.391±0.063 | 103.742±5.907 | 28.087±2.370 |
| | LDM [41] | 0.883±0.090 | 0.310±0.072 | 95.429±4.218 | 19.638±1.294 |
| | Ours | **0.955±0.021** | **0.608±0.033** | **66.729±2.184** | **11.742±0.891** |

*KID is scaled by a factor of 1000

loss, which aids the diffusion process in generating images with greater dispersion within each class and view.

To showcase our proficiency in generating high-resolution images, we propose an analysis with super-resolution methods, LDM [41], and BSRGAN [55]. We begin by employing our pathology diffusion model to generate a $512 \times 512$ overview image, denoted as $I^{ref}$, which is subsequently resized to $2048 \times 2048$. Next, we randomly sample 10, 000 tiles of size $512 \times 512$ from the resized $I^{ref}$

to create a dataset for the upscaling methods. Since the upscaling methods operate on synthetic low-resolution data, it is impractical to compute metrics that require high-resolution ground truth, such as SSIM [50]. Therefore, we utilize previously mentioned similarity metrics and compute the similarity between the generated tiles and real patches from the test set. Table 3 presents the quantitative results of the methods, demonstrating that our model can produce upscaled images with greater similarity to real pathology images.

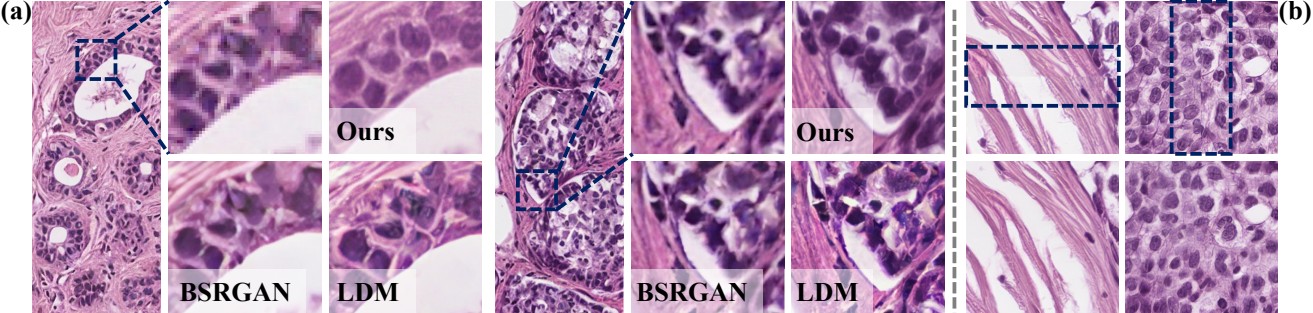

Figure 4: (a) Visualization of inter-model upscale performance. Taking low-resolution reference as a input, our model generates pathology-meaningful spatial context, while others generate artifacts and unrelated detail. (b) Demonstration of tiling artifact refinement. Images tiling artifacts are highlighted by boxes. Our model removes it seamlessly with minor image distortion.

Table 2: Ablation study of our proposed method for Patho-Align. Both 2048 pixels high-resolution and 512 pixel low-resolution generations are measured by IP, IR, FID, KID. PA and FEL are Patho-Align training strategy and Feature Entropy Loss.

|  | Modules | | | | | |
|  | PA | FEL | IP↑ | IR↑ | FID↓ | KID↓ |
|---|---|---|---|---|---|---|
| high-resolution |  |  | $0.906_{\pm0.093}$ | $0.322_{\pm0.084}$ | $83.371_{\pm5.852}$ | $20.795_{\pm2.049}$ |
|  | ✓ |  | $0.959_{\pm0.045}$ | $0.393_{\pm0.063}$ | $59.231_{\pm5.471}$ | $11.173_{\pm1.207}$ |
|  | ✓ | ✓ | $\mathbf{0.964}_{\pm0.012}$ | $\mathbf{0.592}_{\pm0.026}$ | $\mathbf{45.359}_{\pm3.732}$ | $\mathbf{8.139}_{\pm0.413}$ |
| low-resolution |  |  | $0.907_{\pm0.084}$ | $0.324_{\pm0.106}$ | $91.653_{\pm5.932}$ | $9.167_{\pm1.783}$ |
|  | ✓ |  | $0.943_{\pm0.042}$ | $0.417_{\pm0.069}$ | $68.962_{\pm4.510}$ | $11.509_{\pm0.913}$ |
|  | ✓ | ✓ | $\mathbf{0.955}_{\pm0.021}$ | $\mathbf{0.608}_{\pm0.033}$ | $\mathbf{66.729}_{\pm2.184}$ | $\mathbf{10.742}_{\pm0.891}$ |

Table 3: Inter-model upscaling performance comparison.

|  | IP↑ | IR↑ | FID↓ | KID↓ |
|---|---|---|---|---|
| [41] | $0.837_{\pm0.085}$ | $0.376_{\pm0.083}$ | $108.439_{\pm4.316}$ | $17.591_{\pm1.019}$ |
| [55] | $0.645_{\pm0.102}$ | $0.382_{\pm0.097}$ | $153.973_{\pm5.791}$ | $42.920_{\pm3.563}$ |
| Ours | $\mathbf{0.971}_{\pm0.023}$ | $\mathbf{0.633}_{\pm0.035}$ | $\mathbf{57.642}_{\pm1.248}$ | $\mathbf{9.837}_{\pm0.692}$ |

Table 4: Performance comparison for PathUp trained w/ and w/o our proposed weight map $w$ for patch-wise timestep tracking.

|  | IP↑ | IR↑ | FID↓ | KID↓ |
|---|---|---|---|---|
| w/o $w$ | $0.939_{\pm0.034}$ | $0.587_{\pm0.039}$ | $76.278_{\pm3.72}$ | $9.814_{\pm1.036}$ |
| w $w$ | $\mathbf{0.964}_{\pm0.012}$ | $\mathbf{0.592}_{\pm0.026}$ | $\mathbf{45.359}_{\pm3.732}$ | $\mathbf{8.139}_{\pm0.413}$ |

Figure 4(a) provides a visual representation of the upscaled patches generated by various methods, highlighting that our model intricately captures details such as cells, nuclei, and tumor stromas when processing low-resolution images. In contrast, other models either solely sharpen the image or generate non-pathology details.

## 4.5 Ablation Study

We conduct a comprehensive comparison to assess the effectiveness of our proposed modules, evaluating metrics between 10,000 synthetic images of various resolutions and real images from the test dataset. High-resolution images are generated using our proposed patch-wise timestep tracking method. As illustrated in Table 2, our proposed patho-align module achieves significant improvement compared to the model trained without our method, enhancing the generation similarity. This improvement is evident in metrics such as IP, FID, and KID. Furthermore, the integration of the feature entropy loss enhances performance across resolutions, demonstrating its capability to strengthen the generation of high-variety images. The images generated by our model, demonstrated in Fig.3 contain rich spatial context details, encompassing cell nuclei, connective tissue, and tumor stroma. The high similarity with real image patches highlights our model's ability to effectively utilize class-related spatial context to generate various tissue classes. Additionally, our feature entropy loss effectively reduces the variance of performance results. The $\beta$ value selection of our proposed loss is discussed in Fig.5. When $\beta = 0.1$, the image generation quality measured by IR reaches the highest performance. Performance drops when $\beta > 0.1$, which may because optimising our loss affects the descent of $L_r$, weakening the reconstruction ability of model

The effectiveness of tiling artifact removal is assessed by comparing $2048 \times 2048$ images with real patches from the test dataset. Tiling artifacts are introduced by modifying our timestep tracking method to tile latents without incorporating our weight map $w$. The comparison results presented in Table 4 indicate that images refined by our proposed module exhibit superior performance in terms of FID and KID compared to their non-refined counterparts. However, the improvement in IP and IR metrics is marginal, likely due to the limited extent of tiling artifacts within the generated images.

**Table 5: Mean quality score of 3 pathologists on multi-class high-resolution real and synthetic images.**

|  | N | PB | UDH | FEA | ADH | DCIS | IC | Mean |
|---|---|---|---|---|---|---|---|---|
| Real image | $8.167_{\pm0.235}$ | $8.333_{\pm0.249}$ | $7.967_{\pm0.772}$ | $8.500_{\pm0.245}$ | $7.667_{\pm0.330}$ | $7.300_{\pm0.408}$ | $8.633_{\pm0.464}$ | $8.081_{\pm0.363}$ |
| Our synthetic $I^r$ | $8.233_{\pm0.704}$ | $7.533_{\pm0.411}$ | $7.500_{\pm0.779}$ | $7.367_{\pm0.519}$ | $7.533_{\pm0.492}$ | $7.467_{\pm0.624}$ | $8.333_{\pm0.556}$ | $7.709_{\pm0.583}$ |

**Table 6: F-scores on the lesion subtype classification task, comparing models trained with real data only to models trained with random data augmentation (Rand.), and generated lesion images (Ours).**

| Method | N | PB | UDH | FEA | ADH | DCIS | IC | Mean |
|---|---|---|---|---|---|---|---|---|
| [12] | $65.527_{\pm1.077}$ | $43.051_{\pm2.729}$ | $31.149_{\pm1.514}$ | $\mathbf{67.071}_{\pm3.081}$ | $33.297_{\pm1.393}$ | $43.038_{\pm2.417}$ | $69.854_{\pm1.039}$ | $50.427_{\pm1.892}$ |
| [12]+Rand. | $64.489_{\pm2.095}$ | $52.827_{\pm2.713}$ | $33.264_{\pm1.694}$ | $65.215_{\pm2.509}$ | $39.854_{\pm2.058}$ | $45.591_{\pm3.031}$ | $71.457_{\pm1.097}$ | $53.242_{\pm2.171}$ |
| [12]+Ours | $\mathbf{68.041}_{\pm1.048}$ | $\mathbf{55.964}_{\pm2.159}$ | $\mathbf{40.062}_{\pm1.317}$ | $66.834_{\pm2.582}$ | $\mathbf{37.752}_{\pm1.194}$ | $\mathbf{50.015}_{\pm2.261}$ | $\mathbf{72.492}_{\pm0.784}$ | $\mathbf{55.871}_{\pm1.621}$ |
| [25] | $72.473_{\pm1.674}$ | $51.531_{\pm2.409}$ | $38.853_{\pm2.461}$ | $68.801_{\pm2.975}$ | $35.783_{\pm2.475}$ | $52.144_{\pm3.597}$ | $83.314_{\pm1.289}$ | $57.557_{\pm2.554}$ |
| [25]+Rand. | $73.245_{\pm1.211}$ | $53.443_{\pm2.756}$ | $45.136_{\pm3.258}$ | $67.734_{\pm2.781}$ | $43.796_{\pm2.724}$ | $55.037_{\pm2.836}$ | $81.056_{\pm2.071}$ | $59.921_{\pm2.455}$ |
| [25]+Ours | $\mathbf{75.080}_{\pm1.207}$ | $\mathbf{61.425}_{\pm2.047}$ | $\mathbf{51.937}_{\pm2.479}$ | $\mathbf{69.921}_{\pm2.753}$ | $\mathbf{44.675}_{\pm2.078}$ | $\mathbf{57.052}_{\pm2.427}$ | $\mathbf{83.328}_{\pm1.914}$ | $\mathbf{63.345}_{\pm2.129}$ |

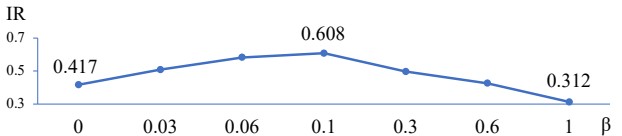

**Figure 5: Synthesis quality measured by IR according to various $\beta$ value. When $\beta = 0.1$, the IR score reaches the best performance.**

Visualizations provided in Figure 4(b) illustrate how our refinement module effectively mitigates tiling artifacts while minimally affecting spatial contexts.

### 4.6 User Study

The pathological plausibility of our synthetic high-resolution images was assessed by three experienced pathologists. For this evaluation, we randomly selected 10 real and synthetic images for each class. The pathologists were instructed to rate the authenticity of each presented pathology image using a quality score ranging from 1 to 10, where 1 indicated "synthetic" and 10 indicated "real." The score across all classes is presented in Table 5. The results demonstrate that our method generates realistic images with a mean quality score of 7.709. Interestingly, our model achieved a higher score than real images in the "N" class.

### 4.7 Downstream Task

We assess the efficacy of our synthetic high-resolution data in a downstream lesion classification task using the BRACS dataset. Our model-generated multi-class high-resolution pathology images serve as augmentation data for training images. We evaluate the performance using two image classification networks, ViT-L [12] and ADMIL [25], for both single-instance and multi-instance learning methods. All high-resolution images are resized to $512 \times 512$ when training ViT-L. Table 6 presents the F-scores comparing the methods trained with and without our augmentation. The results indicate an improvement in performance with augmentation for both single-instance and multi-instance learning models. However, for ViT-L,

the F-score of FEA is lower than that without augmentation. This discrepancy may be attributed to the resizing of high-resolution images, which could lead to a loss of cancer-related spatial context, thereby limiting the classification model's performance.

### 4.8 Limitation and Future Work

While our method exhibits outstanding performance, it is not without limitations. The time-consuming nature of interactive denoising in diffusion models remains a challenge. Furthermore, there is room for optimization to enhance the generalizability of our method across different domains. In the future, efforts will be directed towards reducing the inference time of diffusion models and extending the applicability of our proposed method to diverse datasets.

## 5 CONCLUSION

In summary, we propose a novel generative model for synthesizing multi-resolution lesion subtypes from pathology images. Our method integrates expert pathology knowledge with multi-class images using the patho-align module and an feature entropy loss to enhance inter-class variety in synthetic images. Additionally, we introduce a patch-wise timestep tracking strategy within the latent diffusion model framework to generate high-resolution images and address tiling artifacts using latent weights. Our approach demonstrates effectiveness in generating realistic pathology images across different resolutions and proves useful as a data augmentation method for downstream tasks like lesion subtype classification. Importantly, our focus on modeling multi-resolution spatial context extends beyond data augmentation, paving the way for correlating textural expert knowledge with spatial context.

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
