# OpenReview forum: "PathUp: Patch-wise Timestep Tracking for Multi-class Large Pathology Image Synthesising Diffusion Model"
_acmmm.org/ACMMM/2024/Conference — MM2024 Oral_

### Official Review · Reviewer_hNRo · 2024-05-01

**Rating:** 5
**Confidence:** 4

**Summary:**

The proposed PathUp model introduces a novel diffusion approach specifically designed for synthesizing multi-class high-resolution pathology images, addressing the challenges of high resolution and sparsity in labels. It features a latent space patch-wise timestep tracking to eliminate tiling artifacts and incorporates expert pathology knowledge through the patho-align mechanism. The model's effectiveness is validated through qualitative and quantitative evaluations, including expert reviews, showcasing its potential to enhance downstream tasks like cancer subtype classification through data augmentation.

**Strengths:**

Advanced Image Synthesis Techniques: PathUp employs a novel latent space patch-wise timestep tracking mechanism that effectively produces high-quality synthetic pathology images without the common issue of tiling artifacts. This approach is particularly valuable in handling the high resolution and complexity of whole slide images in pathology, ensuring that the synthetic images are both realistic and free of common synthesis errors.

Integration of Domain Expertise: The model incorporates expert pathology knowledge through the "patho-align" mechanism, which aligns the synthetic image generation process more closely with real-world pathological assessments. This integration ensures that the synthetic images maintain clinical relevance and accuracy, particularly in replicating intricate details necessary for accurate disease diagnosis and classification.

Robust Validation and Utility for Augmentation: PathUp has been rigorously validated through both qualitative and quantitative methods, including assessments by human pathology experts. This comprehensive validation demonstrates the authenticity and high quality of the synthetic images. Moreover, the utility of these images as a tool for data augmentation has been highlighted, showing significant potential to enhance the performance of downstream tasks such as cancer subtype classification, thereby addressing the challenge of sparse labeling in training datasets.

**Limitations:**

Relevance to the Multimedia (MM) Community: The focus of the paper on digital pathology and the synthesis of pathology images may not align closely with the broader interests of the multimedia (MM) community, which often centers on more general image and video processing, analysis, and understanding. This misalignment might limit the interest and applicability of the research within the MM community, potentially affecting its reception and the impact of the findings.

Availability of Code: The paper does not mention whether the code for the PathUp model will be made available upon acceptance. The absence of accessible code can significantly hinder the reproducibility of the research and its adoption by other researchers who might want to replicate the study, validate the findings, or extend the work.

Limited Discussion of Related Works: Although the paper presents an innovative approach, it does not extensively discuss relevant existing literature, such as the "Diffusion-Based Data Augmentation for Nuclei Image Segmentation." This lack of a thorough literature review can be seen as a weakness because it does not adequately situate the research within the context of existing knowledge and advancements. This may lead to missed opportunities for comparing the proposed method with other techniques and fully demonstrating its novelty and improvements over prior approaches.

**Suitability:**

1

---

### Official Review · Reviewer_VL4h · 2024-05-25

**Rating:** 4
**Confidence:** 1

**Summary:**

A method called PathUp is proposed, which is based on a diffusion model and specifically designed for synthesizing multi-class high-resolution pathological images. It models complex spatial context with limited information and seamlessly eliminates tiling artifacts by using a latent space-time step tracking strategy. Specifically, during the training of the diffusion model, an expert pathological alignment mechanism is introduced to integrate expert pathological knowledge into the model. To ensure robust generation of lesion subtypes and scale information, a feature entropy loss function is introduced to increase the inter-class diversity of the synthesized images.

**Strengths:**

1. Claims to have proposed the first generative model to learn the generation of multi-resolution lesion subtypes from pathological images, and demonstrates its effectiveness in downstream tasks.

2. The proposed Patch-wise Timestep Tracking can effectively eliminate the problem of tiling artifacts, which is quite practical.

**Limitations:**

1. Clarifications are needed on the correspondence between cs and images at different scales.

2. How are multi-scale pathological images specifically inputted? Where is the multi-scale aspect reflected in the input?

**Suitability:**

2

---

### Official Review · Reviewer_DUPX · 2024-05-28

**Rating:** 4
**Confidence:** 2

**Summary:**

The paper introduces PathUp, a method for generating multi-class high-resolution pathological images based on the diffusion model. It incorporates Patho-align to integrate expert pathology knowledge with multi-class images and utilizes a patch-wise timestep tracking strategy to address block artifacts. The experimental results validate the effectiveness of the method.

**Strengths:**

(1) The paper is well-written, and the motivation is effectively elucidated.

(2) It integrates expert pathology knowledge to generate multi-class pathological images.

**Limitations:**

(1) The novelty appears somewhat incremental, as similar approaches have been explored in related works, such as incorporating text representations into latent diffusion for generating histopathology images [a] and employing tracking time steps to address tiling artifacts [b]. The paper lacks a thorough description of the current state of research in these areas.
[a] Yellapragada, Srikar, et al. "PathLDM: Text conditioned latent diffusion model for histopathology." WACV 2024
[b] Aversa, Marco, et al. "Diffinfinite: Large mask-image synthesis via parallel random patch diffusion in histopathology." NeurIPS 2024.

(2) The experimental comparison methods appear relatively outdated, as there is a lack of comparisons with current state-of-the-art methods, such as those presented in [20], making it challenging to highlight the effectiveness of the proposed method.

**Suitability:**

2

---

### Meta-Review · Area_Chair_CbLR · 2024-07-02

**Recommendation:** Accept (Oral)
**Confidence:** 5

**Metareview:**

There is a consistent recommendation of acceptance from all reviews.